# Content and Face Validity of the Evaluation Tool of Health Information for Consumers (ETHIC): Getting Health Information Accessible to Patients and Citizens

**DOI:** 10.3390/healthcare11081154

**Published:** 2023-04-17

**Authors:** Simone Cocchi, Chiara Cipolat Mis, Mauro Mazzocut, Irene Barbieri, Maria Chiara Bassi, Silvio Cavuto, Silvia Di Leo, Alessandra Miraglia Raineri, Valentina Cafaro

**Affiliations:** 1Medical Library, Azienda USL—IRCCS di Reggio Emilia, 42123 Reggio Emilia, Italy; 2Biblioteca Scientifica e per Pazienti, Centro di Riferimento Oncologico di Aviano (CRO), IRCCS, 33081 Aviano, Italy; 3Biblioteca di Area Umanistica, Sistema Bibliotecario di Ateneo, Università Ca’ Foscari, 30123 Venice, Italy; 4Psycho-Oncology Unit, Azienda USL—IRCCS di Reggio Emilia, 42123 Reggio Emilia, Italy; 5Clinical Trials and Statistics Unit, SOC Infrastructure, Research and Statistics, Azienda USL—IRCCS di Reggio Emilia, 42123 Reggio Emilia, Italy

**Keywords:** health literacy, patient education, quality information assessment, education materials, health information, validation study

## Abstract

Background: Health information concerns both individuals’ engagement and the way services and professionals provide information to facilitate consumers’ health decision making. Citizens’ and patients’ participation in the management of their own health is related to the availability of tools making health information accessible, thus promoting empowerment and making care more inclusive and fairer. A novel instrument was developed (Evaluation Tool of Health Information for Consumers—ETHIC) for assessing the formal quality of health information materials written in Italian language. This study reports ETHIC’s content and face validity. Methods: A convenience sample of 11 experts and 5 potential users was involved. The former were requested to evaluate relevance and exhaustiveness, the latter both readability and understandability of ETHIC. The Content Validity Index (CVI) was calculated for ETHIC’s sections and items; experts and potential users’ feedback were analyzed by the authors. Results: All sections and most items were evaluated as relevant. A new item was introduced. Potential users provided the researchers with comments that partly confirmed ETHIC’s clarity and understandability. Conclusions: Our findings strongly support the relevance of ETHIC’s sections and items. An updated version of the instrument matching exhaustivity, readability, and understandability criteria was obtained, which will be assessed for further steps of the validation process.

## 1. Introduction

### 1.1. Background

Patient education, patient empowerment, and patient engagement represent pivotal interconnected issues in the healthcare sector. These domains are closely related to the concepts of “Literacy” and “Health Literacy” [1,2,3,4].

Literacy is a set of skills and competences, and not a mere possession of a higher education degree, considered necessary to live adequately and fully in contemporary society [5,6,7]. When this set of skills and abilities is applied to the health sector, we refer to Health Literacy (HL). Literacy and HL are strongly related dimensions. Ratzan and Parker define HL as “the degree to which individuals have the capacity to obtain, process, and understand basic health information and services needed to make appropriate health decisions” [8] (p. vi). A more recent definition of HL tries to highlight other aspects of this complex dimension, pointing out that “Health Literacy is linked to Literacy and entails people’s knowledge, motivation and skills to access, understand, appraise, and apply health information in order to make judgments and take decisions in everyday life concerning healthcare, disease prevention, and health promotion to maintain or improve quality of life during the life course” [9] (p. 3).

Over time, the concept of HL has evolved from the initial focus on individual skills and competences to models taking into account what social and healthcare systems require of individuals [9,10,11,12]. Additionally, HL applies to individuals as well as to healthcare professionals and health systems [13,14]. In particular, “Health Literacy emerges when expectations, preferences, and skills of individuals seeking health information and services meet the expectations, preferences, and skills of those providing information and services” [15] (p. 2). As reported by The Calgary Charter on Health Literacy, “systems can be health literate by providing equal, easy, and shame-free access to and delivery of healthcare and health information” [13] (p. 2).

A link between people’s HL levels and their health status has been documented in the literature. Evidence shows that lower HL levels are related to more hospitalization, inadequate understanding of written health information, lower medication compliance, impaired healthcare knowledge, lower use of preventive services, higher healthcare costs, and, for older people, worse general health status [16].

Research found that a large portion of the population in various OECD (Organization for Economic Co-operation and Development) member countries, as well as in other non-OECD member countries, shows low literacy and numeracy levels, with reference to contemporary society’s demands [7,17,18]. This is highlighted also in the European Health Literacy Survey (HLS-EU) [1], which reported that almost half of the population of the 11 European countries involved had a low level of HL. Additionally, population subgroups disadvantaged due to social, economic, educational, or older age characteristics, frequently showed low levels of HL, “suggesting the presence of a social gradient” [1] (p. 1053), which impact on health equity. In this context, the quality of health information assumes a fundamental role. As stated by Rudd [19] (pp. 1005–1006), “the first call to action requires change in the way health information is offered”, as research has shown that the way many health information resources are set up and written make them suitable only for audiences with high levels of skills [19]. This therefore appears as a paradox of the contemporary era, starting from the Internet revolution onwards, in which an enormous amount of information is available to an equally enormous number of people who may not be able to use it adequately [14].

A key goal for health literate systems and organizations is to develop and deliver good quality written health information materials (i.e., leaflets, brochures, etc.) which are both evidence-based and readable and understandable. This means that content reported within such materials should be designed and formatted in a way that is easily read and understood by consumers with different health literacy levels.

Research reveals that Italy is a low-ranking country with reference to its population literacy and numeracy level. Data from the Survey of Adult Skills [18] show that Italy is 33rd out of 39 countries in terms of population literacy proficiency and 32nd in terms of numeracy proficiency, with about 70% of the population not reaching the level of skills considered necessary to interact effectively in contemporary society.

From November 2011 to March 2013, a group of medical librarians in charge of patient education, health literacy, and health information activities of two cancer research centers in northern Italy developed a tool with a twofold purpose: (a) to assess the formal (i.e., linguistic, textual, graphical, etc.) quality of health information materials written in Italian language; (b) to help healthcare professionals in developing good quality health information materials with specific reference to formal aspects.

### 1.2. Limitations of Already Existing Tools

The need to create a new tool arose from the results of a preliminary literature search conducted on different bibliographic databases (PubMed, Embase, PsycInfo), on national and international health agency websites, and other websites through the Google search engine. Our aim was to identify both existing tools for the evaluation of health information materials and literature about health information quality criteria. All the tools retrieved showed different limitations. Firstly, almost all of the available tools, such as Suitability Assessment of Materials (SAM) [20], Ensuring Quality Information for Patients (EQIP) [21], Consumer Information Rating Form (CIRF) [22], Tool to Evaluate Materials Used in Patient Education (TEMPtEd) [23], and Suitability and Comprehensibility Assessment of Materials (SAM + CAM) [24], have been developed in the context of Anglo-Saxon culture aiming to evaluate health information materials written in English language. Secondly, some tools have been developed to evaluate only some types of health information materials. For example, DISCERN [25], and its Italian version DISCERNere [26], evaluate information materials on treatment choices. Finally, many of these tools included items whose assessment involves a significant rate of subjectivity. For example, different raters could provide different responses to items such as “Vocabulary uses common words”, included in the SAM [20] (p. 51), or “Is the design and layout of the document satisfactory?”, included in the EQIP [21] (p. 171).

The analysis of the existing tools allowed for identifying a set of common domains, such as the understandability of the language, the clarity of the purpose, the correct organization of the text, the typographical clarity, and the comprehensibility of the illustrations. Other domains were taken into consideration by a minority of tools or even by only one, for example, the evidence of recipients’ involvement in the development of the information materials, the presence of the date of production of the documents, and the indication of information sources. These domains were particularly relevant for the purpose of the new tool. Their importance was also supported by the analysis of literature focused on plain language writing and on the writing of clear and simple health information materials [20,27,28,29,30,31], and by the library and information science background of the authors. The new tool, called Evaluation Tool of Health Information for Consumers (ETHIC), was therefore set up to assess the different types of domains described above. ETHIC was designed to be a low-subjectivity tool for medical librarians and health information specialists to evaluate the formal quality of health information materials written in Italian language, and to provide guidance to all health system professionals for drafting simple, clear, reliable, and suitable health information materials for the target audience. ETHIC’s ultimate goal is to have a tangible impact on the quality of health information, hopefully enhancing empowerment and engagement of patients and citizens.

### 1.3. ETHIC’s Main Features

The first version of ETHIC (1.0) was completed in March 2013, integrating features aimed at reducing the rate of subjectivity in the evaluation of some domains. For example, with reference to the evaluation of linguistic aspects, ETHIC assesses readability and lexical comprehensibility by means of two objective parameters: the GULPEASE formula [32] (a specific readability formula for the Italian language) and the quantification of the number of words that belong to the Basic Vocabulary of the Italian language (i.e., that small group of words more easily understood by the majority of the population) [33]. Since manual evaluation of these parameters can be difficult and error-prone, ETHIC 1.0 provided for the use of an external tool for their automated assessment. Furthermore, another external tool was embedded in ETHIC to evaluate the readability of the tables, an element often present in health information materials (PMOSE/IKIRSCH tool [34]).

ETHIC 1.0 consisted of a checklist (24 elements grouped into 5 sections) and the related user manual for evaluating printed documents and their electronic versions. ETHIC 1.0 had been designed to be a modular tool, both to facilitate its adaptation to other languages and cultures (e.g., replacing the language assessment section), and to allow future developments of the tool (e.g., to be applied to web pages, audio–video content, etc.). Furthermore, the scoring system had been designed to allow the evaluation comparison of different types of written information materials. In 2016, ETHIC’s validation was included as part of a larger multicentric research project named “Changing the future: can we effectively improve patient education and its effectiveness in cancer care?”, funded by the Italian Ministry of Health (project code RF-2016-02364211). On this occasion, a new literature search was conducted showing that new tools were available: Health Literacy INDEX [35], Patient Education Materials Assessment Tool (PEMAT) [36], and Centers for Disease Control and Prevention Clear Communication Index [37]. The same process of analysis of both common and relevant domains described above was carried out. Even though these tools showed partially the same limitations as the first tools examined, they still allowed for reconsidering the importance of domains that were not initially included in ETHIC, such as the evaluation of the numerical contents of health information materials. Owing to this process, further modifications were made to the tool that led to a more updated and thorough version of ETHIC (Beta 1, July 2020).

This present article stems from the “Changing the future” project and is aimed at describing ETHIC’s content and face validity process and implementation.

## 2. Materials and Methods

### 2.1. ETHIC Beta 1

The ETHIC Beta 1 version included 27 items grouped into five sections: Transparency, Suitability, Language, Use of Numbers, and Graphical Features.

Transparency (9 items) assesses whether the document allows authors’ identification, understanding when and how contents were written, and whether there are economic or technological barriers that limit the document’s access.

Suitability (5 items) assesses whether the document allows users to understand if it can meet their information needs.

Language (6 items) assesses whether the document has been written in such a way as to ensure that it is understandable to the broadest possible population.

Use of Numbers (3 items) concerns numeracy and assesses how numerical and mathematical information is presented.

Graphical features (4 items) assess the graphical characteristics that make a document clear, legible, and usable.

The user manual describes each section and item, and provides evaluators with instructions on how to calculate and assign scores to single items. 

Each item may be assigned a score of 0, 1, or 2 (in some cases only 0 or 2), depending on the degree to which the health information material possesses the characteristic assessed by that item. For some items, the option “not applicable” (N/A) is also provided. For example, the tables’ readability item will be considered “not applicable” in documents that do not contain tables. By summing each applicable item score, a raw score for every section and a total raw score are obtained. After that, a final score is calculated (per section and total). This final score (per section and total) consists of a percentage computed as the ratio between the raw score obtained and the maximum score obtainable from the document on the basis of the applicable items only. This guarantees the possibility of comparing the evaluations of different types of documents (e.g., leaflets or brochures concerning treatment options, specific diseases, healthcare services access, etc.).

### 2.2. Study Design

This paper reports the first phase of a larger study on the validation process of an assessment instrument, performed in two steps: (1) the evaluation of its content and face validity and (2) the evaluation of its reliability. Content validity was assessed through the CVI index supplemented by feedback from multiple experts in a non-face-to-face approach [38,39]. Face validity was evaluated by asking potential users to complete a questionnaire assessing ETHIC’s clarity and understandability through specific questions for each part of the tool. To the best of our knowledge, face validity was rarely assessed in the validation process of tools aimed at judging the quality of written health information and, when evaluated, it was done jointly with content validity [25,36].

### 2.3. Content Validity

Within our study, we implemented this step by involving a group of experts who were requested to both evaluate the extent to which ETHIC’s items and sections were relevant to the target construct [38], and express their qualified opinion on ETHIC’s contents and structure.

#### 2.3.1. Participants

A multidisciplinary panel of 11 experts from different fields, also including patients’ association representatives, was selected through a convenience sampling to mirror all of ETHIC’s domains. Inclusion criteria were as follows: expertise in the selected areas of interest, i.e., linguistics with plain or sectorial language competencies, health literacy, patient empowerment, education and engagement, health communication, health information documents management. Exclusion criteria included not being previously involved in the development of the tool. The sample comprised four males and seven females. Five experts were between 60 and 70 years old; three were between 40 and 50 years old; the last three, one was between 30 and 40, one between 50 and 60, and other was over 70 years old. They were mainly employed at associations/non-profit organizations of social utility (n = 4), the academic field (n = 3), public health services (n = 3), and a publishing house (n = 1). Three were physicians, two were researchers, two were university professors, and the remaining were a business consultant, an online communication coordinator, an information specialist, and an editor. Health communication/patient education was the most represented area of expertise (n = 6), followed by linguistics (n = 3). Document management of health information materials and counseling concerned two participants.

#### 2.3.2. Data Collection and Assessment Procedures

An ad hoc content validation form was developed and structured into five parts. The first part gathered information on sociodemographic characteristics of participants. In the second part, participants were requested to rate the relevance of each section and item of ETHIC on a 4-point Likert scale (1 = not at all relevant; 4 = very relevant). In the third part, participants could propose further items for existing sections, defining each of them and explaining the rationale supporting their introduction. In the fourth part, participants could suggest further sections other than those just reported in the tool, and items to be included in these sections. Both new sections and items had to be defined, and the rationale supporting their inclusion explained. Finally, participants could report additional comments in the fifth part.

Experts were first contacted by phone by a member of the research group and, if they agreed to participate, they were requested to sign a nondisclosure agreement. As the signed agreement was received by the researchers, the experts were provided by e-mail with the documents to perform the evaluation, i.e., the ETHIC checklist and user manual (Beta 1 version), the content validation form and a cover letter with instructions. Furthermore, in the letter they were informed of the perspective to participate in a remote meeting to be held by the research group and ETHIC’s authors at the end of the content validation process. Subjects had a time window of 6 weeks, between January and April 2021, to complete the evaluation and e-mail the filled-in forms. The remote meeting was held in July 2021 with the purpose of sharing and discussing with the experts’ panel findings and implementation of their evaluation in order to improve the tool.

#### 2.3.3. Data Analysis

Experts’ ratings concerning the relevance of ETHIC sections and items (part 2 of the content validation form) were analyzed by means of the Content Validity Index, recognized in literature as the best practice to quantify CV of an assessment tool [40].

We calculated CVI both for every item (I-CVI) and section of ETHIC (S-CVI). The I-CVI represents “the proportion of content experts giving item a relevance rating of 3 or 4” [39] (p. 52) and it was calculated using the formula I-CVI = (agreed item)/(number of experts). The S-CVI based on the average method (S-CVI) is “the average of the I-CVI scores for all items on the scale or the average of proportion relevance judged by all experts. The proportion relevant is the average of relevance rating by individual expert” [39] (p. 52). S-CVI was calculated using the formula = (sum of I-CVI scores)/(number of items) performed with Microsoft Excel 2019. As Lynn reported [41], an acceptable CVI value is at least 0.78 when the number of panel experts is greater than 9.

Experts’ open responses in the third, fourth, and fifth parts of the validation form were reported in a document, grouped by items and sections. These were first carefully read by each ETHIC author, and then discussed within the authors’ group during a remote meeting. An agreement was reached for every response, in terms of acceptance or rejection, and a decision was made on how to use the response to improve the tool.

A written document addressed to experts involved was developed by the authors’ group, summarizing findings from experts’ evaluations and authors’ decisions about these findings. This document formed the basis for discussion in the remote meeting organized at the end of the content validity implementation.

An updated version of ETHIC, named ETHIC Beta 2 version, was developed according to findings from this phase.

### 2.4. Face Validity

For the purpose of this study phase, a sample of potential users of ETHIC was involved. They were requested to evaluate each part of the tool with reference to its clarity and understandability.

#### 2.4.1. Participants

Five potential users were selected through a convenience sampling. They were recruited starting from the authors’ professional network, as employees with current or recent experience within the Italian National Health System. They worked in four sectors: library in the medical–scientific area, library in the academic field, health communication and training, and public relations and communications. The inclusion criterion was being a potential user of the tool; exclusion criteria were not being previously involved in the development of the tool and not being part of the expert panel previously involved for content validity. The sample comprised four females and one male. Three participants were between 35 and 40 years old, the other two between 41 and 45 years old. They were mainly employed in public health services (n = 4), while only one participant worked in the academic field. Medical–scientific and digital library were the most represented areas of expertise (n = 3), followed by communication in healthcare (n = 2). Two were librarians from medical–scientific areas, one was a librarian from the academic field, one was employed as a hospital manager in the communication and training area, and one as coordinator of a hospital public relations and communications office.

#### 2.4.2. Data Collection and Assessment Procedures

An ad hoc face evaluation form was developed, including two parts. In the first part, information on sociodemographic characteristics of participants was collected; in the second part, participants were requested to evaluate each part of ETHIC (checklist, introduction, description of single sections and items, appendices) with reference to two aspects:Issue 1: Clarity of both the setting and the organization of the text. (a) “In your opinion, is this part of the user manual clear with reference to the layout and organization of the text?”; (b) “What in particular do you think is unclear?”; (c) “For what reason?”.Issue 2: Presence of confusing or difficult words or sentences. (a) “Do you think this part of the user manual contains words or sentences that are difficult to understand?”; (b) “Which words or sentences did you find difficult to understand?”; (c) “What words or sentences would you use as an alternative?”.

Participants were first contacted by phone and, if they agreed to participate, were requested to sign a nondisclosure agreement. As the signed agreement was returned by experts, they were provided by e-mail with the ETHIC Beta 2 version checklist and user manual, the face validation form and a cover letter providing them with instructions. Subjects had a time window of 4 weeks (between January and February 2022) to complete the evaluation and e-mail the filled-in forms.

As for content validity, participants were informed about the perspective to participate in a remote meeting, held in May 2022 by ETHIC’s authors and research group, aimed at sharing and discussing findings from their evaluation.

#### 2.4.3. Data Analysis

In order to assess face validity, we collected and described participants’ answers and summarized them to find shared ideas, opinions, and suggestions to improve ETHIC in terms of clarity and understandability. Participants’ answers were first carefully read by each ETHIC author, and then discussed within the authors’ group. An agreement was reached for every response, in terms of acceptance or rejection, and a decision was made on how to use the response to improve the tool. 

A written document summarizing feedback from participants’ evaluations was prepared by the authors’ group and used to discuss results with contributors during the remote meeting organized at the end of this phase. An updated version of ETHIC, named ETHIC Beta 3 version, was developed according to the findings from this phase.

## 3. Results

### 3.1. Content Validity

Results reported below are presented according to the five parts of the content evaluation form. Parts one and two concern quantitative findings, whereas parts three, four, and five concern experts’ feedback.

#### 3.1.1. Quantitative Findings

All eleven experts accepted to participate in the study and filled in the content validation form.

All ETHIC’s sections showed an acceptable S-CVI value ranging from 0.78 to 1.00: Transparency = 0.78; Graphical Features = 0.91; Suitability = 0.95; Language = 0.99; and Use of Numbers = 1.00. The ETHIC overall CVI value was 0.90. As for the item assessment, 23 out of 27 reported an I-CVI value higher than 0.78, while four of them did not reach the minimum score required. These four items were part of the “Transparency” section.

#### 3.1.2. Experts’ Feedback

In the third part of the content validation form, experts proposed the introduction of four new items referring to the Transparency, Suitability, Language, and Use of Numbers sections. The proposal of new items was rejected in three cases, as the authors evaluated that those were already included in the tool, or that the proposed items were impossible or too difficult to assess by potential users of ETHIC. Only one new item was accepted and integrated in the section Use of Numbers (see Table 1).

In the fourth part, experts suggested two new sections, together with respective items. One, defined “Equity”, refers to the assessment of socially inclusive and not discriminatory contents. The other one, defined “Readability/Legibility”, refers to a comprehensive evaluation of linguistic aspects of documents. Both were rejected by ETHIC’s authors. With reference to the former, although “Equity” took into consideration relevant and actual aspects, ETHIC had not been set up to conduct an evaluation of the contents of information materials. With reference to the latter, it referred to topics already examined in the tool and evaluated by items included in the sections Language and Graphical features. Therefore, the authors did not consider it necessary to create a separate section. However, suggestions provided by the expert who proposed this section allowed the authors to introduce important clarifications throughout different parts of the user manual, and particularly in the introduction.

In the fifth part of the form, a number of additional comments on specific sections and items were reported by experts. They were both positive feedback on ETHIC’s perceived strengths in terms of clarity, usefulness, completeness, and applicability, and suggestions for improvement stemming from ETHIC’s perceived weaknesses. Most suggestions concerned the section Transparency, followed by Language and Suitability. The remaining sections received mostly positive feedback. Of 23 suggestions, 30% were evaluated as suitable by the authors and, consequently, have been used to introduce changes to improve the text (three examples are reported in Table 1).

Other additional comments were not section- or item-specific, but related to general characteristics of the tool, e.g., the focus of ETHIC on the assessment of print documents only. In some cases, the authors valued such comments and used them to clarify assessment procedures or manual instructions. In other cases, comments were evaluated as not suitable, as these concerned aspects that were already included, were beyond the purposes of the tool, or were beyond the purposes of this specific validation phase (an example is reported in Table 1).

A revised version of ETHIC (Beta 2) was obtained from this validation phase, including 28 items grouped in 5 sections.

### 3.2. Face Validity

#### Participants’ Feedback

All five potential users involved for face validity accepted to participate in the study and filled-in the face validation form. Several parts of ETHIC were appraised by participants as clear with reference to both the layout and the organization of the text, and free from difficult/confusing words or phrases; other parts received several comments. Of 40 comments, just over a half of these were evaluated as appropriate by the authors and, consequently, produced changes improving the text.

The introduction was evaluated as clear but very full of concepts, thus the authors accepted the suggestion of participants to introduce changes aimed at improving text readability.

The checklist did not receive comments requesting change, indicating that this part of the tool is clear and understandable for participants.

Concerning the five sections of the user manual, the majority of comments involved the section Transparency, followed by Suitability and Language. Use of Numbers and Graphical features received the lowest number of comments. The final part of the user manual, reporting Appendices, was evaluated by a participant as difficult to understand.

In Table 2, some examples of verbatim participants’ responses provided in the face validation form are reported, together with decisions made by the authors on their appropriateness and the motivations supporting these decisions. 

A revised version of ETHIC (Beta 3) was obtained from this validation phase, including a number of changes that improved both clarity and understandability of the tool, according to participants’ feedback.

## 4. Discussion

### 4.1. Summary of Results

In this paper, we reported the first phase of the validation process of ETHIC, a novel tool focused on the assessment of the formal quality of written health information materials.

Findings from content validity showed that all sections and most items of the tool were evaluated as relevant by involved experts, reaching the cutoff value of 0.78 established for this study according to literature [41]. A minority of items (4 out of 27), all belonging to the Transparency section, did not reach this value, with I-CVI between 0.64 and 0.73. The researchers decided to keep these items in the tool for the following reasons. Firstly, CVI values were slightly below the established cutoff; secondly, only 3 or 4 (depending on item) out of the 11 experts rated these items with a low score (1 or 2 on the 4-point Likert scale). Finally, a slightly higher number of experts, compared to recent literature recommendations (6–10 subjects) [39], was chosen. As a consequence, a lower cutoff was deemed acceptable by the researchers. Experts’ feedback was consistent with quantitative findings, including suggestions and comments that improved the tool. ETHIC was enriched with the introduction of a new item in the section “Use of Numbers”. Several responses from experts allowed the researchers to better describe both assessment procedures and instructions within the user manual. Most changes concerned the Transparency, Suitability, and Language sections.

ETHIC Beta 2 version, obtained from the content validity process, was then evaluated for face validity. Potential users involved in this phase provided the researchers with a number of comments that partly confirmed and partly questioned ETHIC’s clarity and understandability. According to the previous phase, also in face validity, the majority of comments concerned the Transparency, Suitability, and Language sections. The authors implemented an updated version of ETHIC (Beta 3) including a number of changes throughout different parts of the user manual. This version of the tool will be assessed for reliability as a further phase of the validation process.

### 4.2. Strengths and Limitations

A number of strengths and limitations of the study must be considered in order to interpret results. Experts’ and participants’ feedback, concerning both content and face validity, were not anonymized. Moreover, a nonprobability sampling method, namely a convenience sampling, was adopted, comprising respondents who were part of the researchers’ professional network. Both sampling conditions could have influenced responses in terms of relational constraint. Nevertheless, we involved a number of experts deemed as appropriate, or even slightly more than that recommended in the literature [39,41]. Moreover, the choice of participants with a broad and extensive span of expertise provided the researchers with in-depth and detailed feedback and observations.

In this study, we reported a detailed description of both procedures and results of ETHIC’s content and face validity, integrating quantitative data with participants’ feedback. To our knowledge, studies on the validation process of similar tools do not include such detailed description [23,25,36].

Given the innovative features of the tool, the authors’ choice to accept or reject participants’ suggestions was the result of shared decisions based on ETHIC’s specific objectives and characteristics; no information has been retrieved in literature concerning procedures to be implemented in addressing this topic. However, decisions emerged from the in-depth authors’ individual and joint reflections on responses provided by the participants. Furthermore, both criteria and procedures guiding decision-making were made explicit to participants in the two final meetings, which were held to present to them the content and face validity results.

A limitation intrinsic to this paper is that it addresses only the first phase of the entire validation process. Our intention to report in detail both procedures and results of this specific step was driven by the choice to fill the existing literature gap; it concerns the lack of detailed information conveyed by validation studies on health information tools e.g., [21,25,36]. In-depth and transparent reporting of procedures responds to rigorous criteria of dense description of the research method, allowing stepwise replication [42].

### 4.3. Theoretical and Practical Implications

ETHIC aims to assess the quality of health information materials from a formal point of view, taking into consideration several domains not always included in tools described in the literature. Given its specific purposes, it can be employed to evaluate a broad variety of documents, notwithstanding the topics they address.

It was initially developed to overcome the limitations of existing tools, some of which are dependent on the context they stem from. Likewise, ETHIC’s characteristics reflect Italian language and culture; nevertheless, its modular structure can support both the development of tools with the same objectives in diverse linguistic and cultural backgrounds, and the development of tools for the assessment of other kinds of health information materials, such as web pages, audio–video materials, etc. 

Moreover, since it was meant for developing and evaluating written health information materials with reference to readability and understandability, it represents an advance in the health information and patient education fields. From an ecological perspective [43], health information focuses not only on patients and their ability to manage their own health—it also concerns the way services and professionals provide information to patients, family members, and communities in order to ensure an equitable redistribution of community resources. Thus, promoting health information and patient education implies acting jointly within a context consisting of multiple actors and systems [44,45]. Health professionals and services are some of those systems, whose role is providing people and communities with the tools to be able to make informed decisions about their health, promoting empowerment [46,47]. Starting from this perspective, tools focused on making health information accessible, as ETHIC does, represents a means to make care and health more inclusive and fairer [48].

## 5. Conclusions

This work focuses on the validation process of a tool aimed at improving patient and healthcare system empowerment, making health information accessible. This represents one of the conditions that allow citizens and patients to actively participate in the management of their own health. The development of ETHIC is supposed to be a starting point towards a more inclusive healthcare system.

Our findings strongly support the relevance of ETHIC’s sections and items with reference to its purpose. The amount and wealth of information collected and analyzed led to a deep and multifaceted process to improve the tool. Input provided by both experts and potential users allowed us to obtain an updated version of the tool matching exhaustivity, readability, and understandability criteria. In order to proceed with the validation process, the ETHIC Beta 3 version will be assessed for reliability. This further step will test the degree of accuracy, objectivity, and applicability of the tool.

## Figures and Tables

**Table 1 healthcare-11-01154-t001:** Examples of experts’ feedback from different sections of the content validation form, and decisions made by authors.

Section	Experts’ Feedback	Authors’ Decision	Authors’ Motivations
Transparency	“[…] Accessibility is a very broad term that includes many aspects. Contemplating it as an item within the ‘Transparency’ section is perhaps reductive. Here it is meant as ease of access in the sense of availability […] ‘Accessibility’ refers to the possibility of accessing a text or information even by those who use assistive technologies, or particular devices, and it is a requirement that is not easy to test and verify.”	Accepted	The item is intended to evaluate whether the resource is free and freely accessible without technological constraints. We will replace the item “Accessibility of the document” with “Availability of the document”
Language	“[...] Within individual categories, several parameters are sometimes listed, and the evaluation is based on the number of parameters met. However, the parameters listed do not all have the same rank. For example, parameter 4 (prevalence of sentences with only one main piece of information) is much more significant than parameter 1 (subject explained in each sentence) or 5 (absence of sentences including lists)” [...]	Rejected	This is an acceptable compromise between the depth of the evaluations and the sustainability of the evaluation process itself, which should be carried out by librarians and health information professionals and not by linguists or technicians. Some of the parameters examined may not have the same rank from a strictly linguistic point of view, but could have the same impact on the potential reader thinking from the perspective of health literacy principles and practices
Graphical Features	“The evaluation of the tables possibly present in the document appears a bit difficult and hardly applicable in practice”	Rejected	The evaluation of the tables can be difficult to perform. This aspect will be assessed in the next phase of the validation process (reliability).
Use of Numbers	“I suggest inserting the item ‘Quantification of risk with absolute rather than relative numbers’”	Accepted	This item takes into consideration a relevant aspect that is not currently covered by the tool.
Additional comments	“It is not clear to me if ETHIC also applies to web/web-only tools.” “I wonder if the inclusion of videos (linked from the written text, obviously in the case of Internet resources) is possibly pertinent to this manual…”	Rejected	The comments received were very valuable, because they allowed us to understand that this specification is not sufficiently evident either on the cover or in the text of the manual. Therefore, we will make the intended use of ETHIC more explicit, which in this version subject to validation is confirmed to be reserved only for printed texts (booklets, brochures, and the like) and their electronic versions

**Table 2 healthcare-11-01154-t002:** Examples of participants’ feedback to issues 1 and 2 of the face validation form, and decisions made by the authors.

Section/Paragraph/Part	Participants’ Feedback	Authors’ Decision	Authors’ Motivations
Transparency	Issue 1 “The indistinct use of the term ‘author’ and ‘responsible’. I would always use ‘responsible’ rather than author, as this term is reserved for a specific kind of responsibility”.	Accepted	The terms related to authorship will be replaced with the expression “responsible for the contents”, and all the text will be modified accordingly.
Suitability	Issue 2 “’Hierarchization’ could be replaced by ‘order and importance’”.	Rejected	The meaning of “hierarchization” is explained in the text
Suitability	Issue 1 “[…] there is a sentence that is too long, probably due to the translation of a text (it is quoted): I don’t know how to handle this problem, I don’t know the original text from which I assume it comes from”.	Accepted	The sentence will be split in two parts to improve readability.
Language	Issue 1 “If I need to evaluate documents without verbs, how should I proceed? E.g., in brochures with noun phrases, should I deduce and evaluate the verb? How can I proceed with documents with only images whose texts consist only of captions? I cannot find this information in the manual”.	Accepted	This item is aimed at evaluating the kinds of verbs used in the document. ETHIC provides users with information to assess also noun phrases that contain unconjugated verbs. Although phrases without verbs are not frequent in documents assessed by ETHIC, we will add some information to the text to highlight this specific case.
Graphical Features	Issue 1 “The topic of accepted and unaccepted graphical devices. For example, italics, used consistently throughout the text with a value other than the conventional/expected in the manual, is not acceptable? In my opinion, the aspect that needs to be evaluated is the consistency of use and not adherence to a convention to which the writer may not adhere for valid reasons”.	Rejected	This item already considers the evaluation of the correct and coherent use of typographical devices. In the manual, some information is reported on how italics should be used; nevertheless, any other use of italics is considered appropriate if consistent with information provided in the other parts of the manual.
Appendices	Issue 1 “The concepts illustrated in the evaluation of the tables require a greater level of understanding than the previous ones”.	Rejected	As difficulty related to this part is due to characteristics intrinsic to the ancillary tools that need to be employed for evaluating the legibility tables, we will not change the current version of the text.

## Data Availability

The data that support the findings of this study are available from the corresponding authors (I.B., S.D.L.) upon reasonable request.

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
