# Peer review of "Content and Face Validity of the Evaluation Tool of Health Information for Consumers (ETHIC): Getting Health Information Accessible to Patients and Citizens"

_healthcare, 2023, doi:10.3390/healthcare11081154_

Round 1

Reviewer 1 Report

The article analyzes the topic is very relevant in the modern context of public information. The authors definitely showed the relevance of the studya nd highlighted the problem. The research data presented in a detailed and structured manner. The conclusions, limitations and strengths of the study are clearly presented.

I would like to make a few comments:

a)       in the theoretical part it is indicated that: "...in various OECD (Organization for Economic Co-operation and Development) countries shows low literacy and numeracy levels, with reference to contemporary society's demands" (p. 71-73), but the presented data are quite old sources. Perhaps more recent sources should be provided;

b)      p.p. 96-100  - while reading the text, it remains unclear whether the tools have been completed and whether it have been used;

c)       p. 258 - please clarify the statement "...and being part of the authors' professional network", it is not very clear about what kind of professional network are writing here;

d)      p.p. 299-309 and p.p. 360-368 - in my opinion, it would be more appropriate to present this information in the Participants sections 2.3.1 and 2.4.1.

Author Response

The article analyzes the topic is very relevant in the modern context of public information. The authors definitely showed the relevance of the study and highlighted the problem. The research data presented in a detailed and structured manner. The conclusions, limitations and strengths of the study are clearly presented.

We thank the reviewer for appreciating our study. We carefully considered all the comments provided by the reviewer with the aim of improving the present article.

In the theoretical part it is indicated that: "...in various OECD (Organization for Economic Co-operation and Development) countries shows low literacy and numeracy levels, with reference to contemporary society's demands" (p. 71-73), but the presented data are quite old sources. Perhaps more recent sources should be provided;

According to the reviewer’s suggestion, we reported data obtained from the more recent resource available, covering not only OECD member countries,  but also some non-OECD countries. Therefore we added a new reference and, as a consequence, the bibliography has been updated and the reference numbers from 18 to 47 has changed to 19-48.

p.p. 96-100 - while reading the text, it remains unclear whether the tools have been completed and whether it have been used;

We slightly modified the sentence by replacing the expression “began developing” with the term “developed” and clarifying the period of development. In this sentence we refer to the development of the first version of the tool. ETHIC has not yet been employed by its potential users, as we plan to authorize its employment once the validation process is completed.

p. 258 - please clarify the statement "...and being part of the authors' professional network", it is not very clear about what kind of professional network are writing here;

We modified the statement in order to clarify the characteristics of both the sample and the sampling method. We replaced “...and being part of the authors’ professional network” with They were recruited starting from the authors’ professional network, as employees with current or recent experience within the Italian National Health System. We also did slight changes in section 4.2 Strengths and Limitationsconcerning the characteristics of the sample.

p.p. 299-309 and p.p. 360-368 - in my opinion, it would be more appropriate to present this information in the Participants sections 2.3.1 and 2.4.1.

We thank the reviewer for this suggestion. Consequently, we moved information on participants characteristics from 3.1. and 3.2.1 to  2.3.1 and 2.4.1 paragraphs, respectively.

Reviewer 2 Report

The convenience sample of Five participants is very small and limits external validity. The sample includes academics and health professionals, but there is no mentions of community representation or patients. The inclusion of these people is critical, given they will most likely be the end-user of the information.

A summary of what the key topics identified by experts were, would be a great addition on what those actual topics were.

Also, a list of what the key topics identified by experts should be listed out and included.

In the "Limitations of Existing Tools" section, line 108 states that different limitations were found, starting with the availability developed in Anglo-Saxon culture. 

What were some of the other key topics identified from this search? It seems that only a few were described, but were there others? Also, how does the new ETHIC tool start from existing tools? Did they adapt items from previous measures found in the literature search? Did they identify content areas from previous measures that were missing to improve upon in ETHIC? And most importantly, what were the cultural adaptations made for a measure developed in Italy? How was it different culturally, from prior Anglo-Saxon tools?

Author Response

English language and style are fine/minor spell check required.

According to the reviewer’s suggestions, we revised English language and corrected some minor typos and syntax errors throughout the paper.

The convenience sample of Five participants is very small and limits external validity. The sample includes academics and health professionals, but there is no mentions of community representation or patients. The inclusion of these people is critical, given they will most likely be the end-user of the information.

Although it is very small, the amount of the convenience sample involved in the face validity of ETHIC is both consistent with the objective of this validation phase (i.e. assessing  the clarity and the understandability of the tool) and with information retrieved in literature on the implementation of face validity in tools with similar characteristics  (see for  example: Shoemaker, Wolf & Brach 2014 on PEMAT development and validation).

Concerning participants’ background, we chose not to involve patients or representatives from patients’ associations in this phase. Although they are the end-user of the information, they were not meant as the potential users of ETHIC. Indeed, the tool, was developed to be employed by specific kinds of professionals in the health information field, Representatives from patient associations were involved as experts in the content validity, given its focus on evaluating relevance and exhaustivity of the sections and items of the tool.

A summary of what the key topics identified by experts were, would be a great addition on what those actual topics were. 

Also, a list of what the key topics identified by experts should be listed out and included.

In the "Limitations of Existing Tools" section, line 108 states that different limitations were found, starting with the availability developed in Anglo-Saxon culture. 

What were some of the other key topics identified from this search? It seems that only a few were described, but were there others? Also, how does the new ETHIC tool start from existing tools? Did they adapt items from previous measures found in the literature search? Did they identify content areas from previous measures that were missing to improve upon in ETHIC? And most importantly, what were the cultural adaptations made for a measure developed in Italy? How was it different culturally, from prior Anglo-Saxon tools?

We are very grateful to the reviewer for this valuable observation, which allows us to better explain the development of ETHIC. An example has been added concerning the limits of existing tools. Besides, we added some information within paragraphs 1.2 Limitations of Already Existing Tools and 1.3 ETHIC's Main Features (the latter was erroneously indicated in the first version of the paper as 1.2). Here we explained more in detail the development of ETHIC based on the domains found in the literature, as well as the focus on Italian language.

Reviewer 3 Report

Abstract - The background does not elucidate well on the relevance of the study. Perhaps rewrite it more clearly, relating it to the objective of the study

Line 123 - for greater clarity, should be added "in Italian language"

Line 234-235 - authors used "proportion relevance" and "proportion relevant"; the same terminology should be used

Line 244 - "was developed" instead of "has been developed" since all the rest of the text is in the past tense

Line 258 - Why being part of the authors' network was an inclusion criteria? it was important for the study? Or was just convenient to authors? If it's the latter, probably you shouldn't consider it as an inclusion criteria...

Line 289 - "was prepared" instead of "has been prepared" since all the rest of the text is in the past tense

Line 301-302 - suggested change of wording: "the last three, one was between 30 and 40, one between 50 and 60 and other was over 70 years old"

Line 302 - the word "respectively" is unnecessary

Line 309 - the word "respectively" is unnecessary

Line 310 - redundant; maybe just the reference of the range of values and, if you want to show values acceptability, the reference nunber could be shown in parenthesis

Line 316 - last sentence: what do you mean here? that these items were part of this section? rewrite for clarity

Line 319-321 - Perhaps examples of these items (both the proposed and existing ones) could be provided. Or are these items included in table 1? If so, please mention it in the text

Line 321 - the word "respectively" is unnecessary

Line 342-344 - These suggestions are shown at table 1? Please mention it in the text

Line 420 - In this section of the discussion, I missed seeing a comparison between the instrument resulting from the face and content validation process and existing instruments on the same topic, namely, showing its superiority over those. Not only the content could be discussed in the light of the improvements this tool aims to introduce, when compared to existing ones, but also the decisions to accept/reject the experts' or users' suggestions could be explained in the same way.

Author Response

English language and style are fine/minor spell check required.

According to the reviewer’s suggestions, we revised English language and corrected some minor typos and syntax errors throughout the paper.

Abstract - The background does not elucidate well on the relevance of the study. Perhaps rewrite it more clearly, relating it to the objective of the study.

According to the reviewer’ suggestion, we changed the first part of the abstract by making reference to the domains of health information and empowerment, closely related to the added value of the tool presented in the study. We hope this change can improve the background.

Line 123 - for greater clarity, should be added "in Italian language".

We clarified the sentence by adding the suggested expression “in Italian language”. 

Line 234-235 - authors used "proportion relevance" and "proportion relevant"; the same terminology should be used.

We thank the reviewer for this suggestion. Unfortunately, we cannot change the text because in this part of the paragraph a literal quotation is reported (see reference n. 39, p.52).

Line 244 - "was developed" instead of "has been developed" since all the rest of the text is in the past tense.

According to the reviewer’s suggestion, we replaced the expression “has been developed” with “was developed.

Line 258 - Why being part of the authors' network was an inclusion criteria? it was important for the study? Or was just convenient to authors? If it's the latter, probably you shouldn't consider it as an inclusion criteria...

We thank the reviewer for this comment. We agree with the reviewer that being part of the authors’ network is not an inclusion criteria, but a characteristic intrinsic to the convenience sampling method chosen for the study. As a consequence, we modified the sentence accordingly.

Line 289 - "was prepared" instead of "has been prepared" since all the rest of the text is in the past tense.

We replaced the expression “has been prepared” with the expression “was prepared.

Line 301-302 - suggested change of wording: "the last three, one was between 30 and 40, one between 50 and 60 and other was over 70 years old".

Thank you for your suggestion. We modified the sentence accordingly.  The sentence has been moved to section 2.3.1, as requested by reviewer 1.

Line 302 - the word "respectively" is unnecessary.

We deleted the word “respectively” from the sentence.

Line 309 - the word "respectively" is unnecessary.

We deleted the word “respectively” from the sentence.

Line 310 - redundant; maybe just the reference of the range of values and, if you want to show values acceptability, the reference number could be shown in parenthesis.

Thank you for your suggestion. We modified the sentence, and we deleted redundant information. The S-CVI values of each section have been listed.

Line 316 - last sentence: what do you mean here? that these items were part of this section? rewrite for clarity.

We meant that all four items were included in the transparency” section. We clarified the sentence accordingly.

Line 319-321 - Perhaps examples of these items (both the proposed and existing ones) could be provided. Or are these items included in table 1? If so, please mention it in the text.

One example of these items is reported in table 1. As a consequence, we added a reference to the table in the text. The remaining three items are mentioned in the following part of the paragraph, as these are included in the new sections proposed by experts, named “Equity” and “Readability/Legibility”.

Line 321 - the word "respectively" is unnecessary.

We deleted the word “respectively” from the sentence.

Line 342-344 - These suggestions are shown at table 1? Please mention it in the text.

One example of additional comment is reported in table 1. We added a reference to the table in the text.

Line 420 - In this section of the discussion, I missed seeing a comparison between the instrument resulting from the face and content validation process and existing instruments on the same topic, namely, showing its superiority over those. Not only the content could be discussed in the light of the improvements this tool aims to introduce, when compared to existing ones, but also the decisions to accept/reject the experts' or users' suggestions could be explained in the same way.

We thank the reviewer for this suggestion. We added the integration required within the paragraphStrengths and Limitations, which already includes information on methodology and on the procedure implemented for accepting/rejecting participants suggestions.